# Anodic Stripping Voltammetric Analysis of Trace Arsenic(III) on a Au-Stained Au Nanoparticles/Pyridine/Carboxylated Multiwalled Carbon Nanotubes/Glassy Carbon Electrode

**DOI:** 10.3390/nano12091450

**Published:** 2022-04-24

**Authors:** Yun Du, Chenglong Sun, Yuru Shen, Luyao Liu, Mingjian Chen, Qingji Xie, Hongbo Xiao

**Affiliations:** 1Key Laboratory of Chemical Biology & Traditional Chinese Medicine Research, Ministry of Education of China, College of Chemistry and Chemical Engineering, Hunan Normal University, Changsha 410081, China; hello_dudu1990@163.com (Y.D.); 15588654717@163.com (C.S.); shenyuru163@163.com (Y.S.); lly19970214@163.com (L.L.); mingjian412@163.com (M.C.); 2Changsha Center for Diseases Prevention and Control, Changsha 410004, China; 3College of Science, Central South University of Forestry and Technology, Changsha 410004, China

**Keywords:** pyridine, carboxylated multiwalled carbon nanotubes, gold nanoparticles, gold staining, linear sweep anodic stripping voltammetry, As(III) analysis

## Abstract

A Au-stained Au nanoparticle (Au_s_)/pyridine (Py)/carboxylated multiwalled carbon nanotubes (C-MWCNTs)/glassy carbon electrode (GCE) was prepared for the sensitive analysis of As(III) by cast-coating of C-MWCNTs on a GCE, electroreduction of 4-cyanopyridine (cPy) to Py, adsorption of gold nanoparticles (AuNPs), and gold staining. The Py/C-MWCNTs/GCE can provide abundant active surface sites for the stable loading of AuNPs and then the AuNPs-initiated Au staining in HAuCl_4_ + NH_2_OH solution, giving a large surface area of Au on the Au_s_/Py/C-MWCNTs/GCE for the linear sweep anodic stripping voltammetry (LSASV) analysis of As(III). At a high potential-sweep rate of 5 V s^−1^, sharp two-step oxidation peaks of As(0) to As(III) and As(III) to As(V) were obtained to realize the sensitive dual-signal detection of As(III). Under optimal conditions, the ASLSV peak currents for oxidation of As(0) to As(III) and of As(III) to As(V) are linear with a concentration of As(III) from 0.01 to 8 μM with a sensitivity of 0.741 mA μM^−1^ and a limit of detection (LOD) of 3.3 nM (0.25 ppb) (*S*/*N* = 3), and from 0.01 to 8.0 μM with a sensitivity of 0.175 mA μM^−1^ and an LOD of 16.7 nM (1.20 ppb) (*S*/*N* = 3), respectively. Determination of As(III) in real water samples yielded satisfactory results.

## 1. Introduction

Arsenic is widely distributed in the environment and poses a threat to human health [1,2,3,4]. Among inorganic arsenic compounds, As(III) has a higher toxicity than As(V) [5,6]. According to the guidelines issued by the World Health Organization, the amount of arsenic should not be higher than 10 ppb in drinking water. On the other hand, an ultralow dose of arsenic trioxide, also known as a Chinese traditional medicine, can well treat acute promyelocytic leukemia [7]. Therefore, the rapid and sensitive detection of As(III) is interesting and important. To date, the analytical methods for arsenic detection mainly include inductively coupled plasma mass spectrometry (ICP-MS) [8], atomic absorption spectrometry (AAS) [8], atomic fluorescence spectrometry (AFS) [8], fluorescent probe method [9,10], biosensing [11], and electroanalysis [12], each with its own characteristics.

The electroanalysis methods, especially anodic stripping voltammetry (ASV), have the advantages of being low cost and using portable instruments, making them appropriate for the rapid and in-field detection of arsenic [13,14,15,16]. The ASV analysis of As(III) involves three steps: (1) preconcentration of As(0) under solution-stirred conditions by cathodic potentiostatic reduction of As(III); (2) ceasing solution-stirring to achieve minimum mass transfer in the bulk solution and minimum potentiostatic electrolysis from the stationary bulk solution; and (3) oxidation of As(0) to As(III) and then to As(V) by an anodic potential sweep. Usually, Au electrodes and Pt electrodes, as well as their modified forms, are used for the ASV analysis of As(III) [13]. Pt has a rather strong affinity for As(0) [17,18,19,20], and thus the anodic stripping of As(0) is somewhat difficult on a Pt electrode, making the ASV peak current rather low and the current peak rather broad. In contrast, as Au has an appropriate affinity for As(0), the anodic stripping of As(0) is easy and gives a high ASV peak current and a sharp current peak on a Au electrode, and thus the Au electrode is more favorable for the sensitive analysis of As(III). The Au-involved chemically modified electrodes, e.g., the electrodes modified with gold nanoparticles (AuNPs) [21,22] or their composites with metal oxides [23], silicon materials [24], carbon materials [25,26], and appropriate organic ligands [27,28,29], can further improve the arsenic analysis performance, either by stabilizing the modified Au material or increasing the effective area of exposed Au surfaces. The pyridine-like organics can act as excellent gold ligands to support and stabilize AuNPs [30,31] and may be developed as new materials for high-performance Au-based arsenic analysis.

Herein, we report the preparation of a Au-stained Au nanoparticle (Au_s_)/pyridine (Py)/carboxylated multiwalled carbon nanotubes (C-MWCNTs)/glassy carbon electrode (GCE) by cast-coating of C-MWCNTs on a GCE, electroreduction of 4-cyanopyridine (cPy) to Py, adsorption of gold nanoparticles (AuNPs), and gold staining. The use of 4-cyanopyridine (Py) can lead to the cleavage of the C-C bond between the pyridyl and the cyano group when the 4-cyanopyridine undergoes electroreduction on the electrode, resulting in a pyridine radical. The pyridine radicals are highly active and can be covalently bound well onto the surface of C-MWCNTs/GCE to form Py/C-MWCNTs/GCE. The Py/C-MWCNTs/GCE can provide a good substrate for AuNPs loading and gold staining to increase the Au surface area. Under optimized conditions, the Au_s_/Py/C-MWCNTs/GCE can be used for the ASV analysis of As(III), with high sensitivity, low detection limit, high selectivity, good stability, and reproducibility.

## 2. Experimental Section

### 2.1. Instrumentation and Reagents

Electrochemical measurements were performed on a CHI660E electrochemical workstation with a three-electrode system. A disk GCE (3.0 mm diameter, 0.0707 cm^2^ geometric area) and its modified electrodes served as the working electrode, a KCl-saturated calomel electrode (SCE) served as the reference electrode, and a platinum wire served as the auxiliary electrode. All potentials are reported versus SCE. A PHS-3C pH meter (Leici, Shanghai, China) was used for pH measurements. The surface plasmon resonance absorption spectra of AuNPs were collected on a UV–visible spectrometer (UV-2450, Shimadzu, Japan). X-ray photoelectron spectroscopy (XPS) data were collected on a Thermo ESCALAB 250XI. A field emission scanning electron microscope (MIRA3 LMH, TESCAN) was used to acquire scanning electron microscopy (SEM) images, and it was equipped with a MAX20 energy-dispersive X-ray spectroscopy (EDX) system for elemental analysis.

4-Cyanopyridine, As(III) stock solution, and HAuCl_4_·3H_2_O were commercially obtained from J&K Scientific (Nanjing, China). Carboxylated multiwalled carbon nanotubes were commercially obtained from XFNANO, Inc. (Nanjing, China). Anhydrous ethanol, K_4_Fe(CN)_6_·3H_2_O, K_2_SO_4_, trisodium citrate, and CuSO_4_·5H_2_O were commercially obtained from Chemicals Company of Tianjin (Tianjin, China). All chemicals are of analytical grade or higher quality. A stock solution of 1.0 mM As(III) in 0.5 M aqueous H_2_SO_4_ was prepared and stored in a refrigerator at 4 °C, and a series of As(III) standard solutions at the desired concentrations were prepared for immediate use by diluting this As(III) stock solution with 0.5 M aqueous H_2_SO_4_. Milli-Q ultrapure water (Millipore, ≥18 MΩ cm, USA) was used throughout. The experiments were conducted at room temperature (ca. 25 °C).

### 2.2. Preparation of Au_s_/Py/C-MWCNTs/GCE

The bare GCE, after being physically polished, chemically cleaned, and electrochemically cleaned, was characterized in 0.1 M K_2_SO_4_ solution containing 2.0 mM K_4_[Fe(CN)_6_] to ensure the high electrode activity.

The cleaned GCE was cast-coated with 6 μL of 0.2 mg/mL C-MWCNTs dispersion and air-dried to obtain C-MWCNTs/GCE. The C-MWCNTs/GCE was then placed in 0.1 M aqueous H_2_SO_4_ containing 10 mM 4-cyanopyridine (cPy) and subjected to cyclic voltammetry (CV) treatment for 3 cycles (−1.2~−0.5 V, 0.1 V s^−1^) to obtain Py/C-MWCNTs/GCE. After being water-rinsed and air-dried, the Py/C-MWCNTs/GCE was cast-coated with 10 μL of AuNPs dispersion for 20 min, washed with ultrapure water, and dried with N_2_ to obtain AuNPs/Py/C-MWCNTs/GCE. The AuNPs/Py/C-MWCNTs/GCE was again cast-coated with 10 μL of gold staining solution (3 mM HAuCl_4_ + 18 mM NH_2_OH·HCl) and, after standing for 5 min, washed with ultrapure water to obtain Au_s_/Py/C-MWCNTs/GCE. The electrode preparation is shown in Figure 1.

### 2.3. Electroanalysis of Arsenic(III)

As(III) was detected by linear sweep anodic stripping voltammetry (LSASV). The working electrode was immersed in a stirred 0.1 M aqueous H_2_SO_4_ containing As(III) at a desired concentration to enrich As(0) at −0.40 V for 420 s. The solution-stirring was then stopped for 15 s, and then the linear sweep anodic stripping of As(0) at 5 V s^−1^ was conducted from −0.40 V to 1.15 V.

## 3. Results and Discussion

### 3.1. Preparation and Characterization of Au_s_/Py/C-MWCNTs/GCE

The AuNPs were prepared according to Ji et al. [32]. The UV-Vis absorption spectrum shows a surface plasmon resonance absorption peak of AuNPs at 521 nm, as shown in Appendix A. The electroreduction of cPy at different pH values was investigated by CV. As shown in Figure 1, 10 mM cPy showed irreversible electroreduction signals at pH 1.0, 3.0, 7.0, 11.0, and 13.0. With the increase in solution pH, the reduction peak shifted negatively and the intensity was weakened. Almost no reduction peak was found in the alkaline environment in the examined potential range. Hence, the electroreduction reaction of cPy was an electron transfer reaction coupled with proton transfer [33,34]. Finally, 0.1 M aqueous H_2_SO_4_ was selected to dissolve cPy for its electroreduction. As shown in Figure 1B, the irreversible reduction peak at ca. −0.8 V resulted from the reduction of cPy [35]. The electroreduction of cPy will reach saturation in a short time, forming a saturated thin layer structure. In the electrode preparation, the number of cPy-electroreduction cycles was selected to be 3.

C-MWCNTs/GCE and Py/C-MWCNTs/GCE were electrochemically characterized with redox probe 1.0 mM K_4_Fe(CN)_6_ in 0.1 M phosphate buffer solution (PBS) at pH 7.0, as shown in Appendix A. Compared with GCE, C-MWCNTs/GCE also showed reversible redox peaks, and the peak currents became slightly larger, indicating good electrode activity. The Py/C-MWCNTs/GCE showed a slightly decreased electrochemical activity versus C-MWCNTs/GCE, implying that an electron-insulating thin-layer Py has been bonded to the electrode surface.

The modified materials and electrodes were characterized by XPS, as shown in Figure 2. After the N1s peaks of Py are separated, the peak positions of Py N and cyano N are at 399.0 eV and 399.8 eV, respectively. When the atoms in the ligand are coordinated or protonated, the density of the electron cloud decreases to a certain extent, which is reflected in the positive shift of the peak position in the XPS spectrum [36,37]. The N1s peak positions of Py and protonated N are at 399.9 eV and 401.9 eV, respectively, and both peaks have a certain degree of positive shift, indicating that the Py ligand was successfully modified on the C-MWCNTs/GCE. Obvious Au 4f peaks are found in Figure 2A,D, indicating that the Au_s_/Py/C-MWCNTs/GCE has been successfully prepared.

The Au_s_/Py/C-MWCNTs/GCE was characterized by CV in 0.1 M aqueous H_2_SO_4_, as shown in Figure 3. The anodic peaks of ca. 1.1 V and 1.5 V are assigned to the formation peaks of gold oxides (AuO_x_). The cathodic peaks of ca. 0 V and 0.9 V are assigned to the reduction peaks of H^+^ and AuO_x_, respectively. As shown in Appendix A, with the increase in the gold-staining time, the current intensity of the reduction peak of AuO_x_ increased significantly, and the increase slowed down after the time exceeded 5 min. Therefore, the gold-staining time of 5 min was used in the subsequent experiments. The real surface area of Au (*S*_Au_) is positively correlated with the charge of the reduction peak of AuO_x_ ($Q_{{\mathrm{AuO}}_{\text{x}}}$) [38] and can be estimated using a conversion factor of 390 μC cm^−2^. The ratio of *S*_Au_ to the geometrical area of the electrode (0.0707 cm^2^) gives the roughness factor (*R*_f_). Next, the effect of the time of gold dyeing on the roughness *R*_f_ was investigated. Reduction peaks of AuO_x_ were observed for Au_/GCE_, Au_s_/C-MWCNTs/GCE, Au_s_/Py/GCE, and Au_s_/Py/C-MWCNTs/GCE, but the heights of reduction peaks are obviously different. The $Q_{{\mathrm{AuO}}_{\text{x}}}$ of Au_s_/Py/C-MWCNTs/GCE is 58.2 μC, corresponding to *S*_Au_ = 0.149 cm^−2^ and *R*_f_ = 2.12, which are significantly higher than those of Au_/GCE_, Au_s_/C-MWCNTs/GCE, and Au_s_/Py/GCE, indicating that gold staining is a simple and effective method to increase *R*_f_.

The modified electrodes were characterized by SEM and EDX, as shown in Figure 4. The Py/GCE shows a rather smooth surface. The Py/C-MWCNTs/GCE shows random stacking of obvious nanotubes (C-MWCNTs). The AuNPs/Py/C-MWCNTs/GCE shows obvious and uniform distribution of AuNPs on the GCE substrate and C-MWCNTs. The AuNPs/Py/C-MWCNTs/GCE shows an increased size and number of Au_s_. N and Au elements are found on the AuNPs/Py/C-MWCNTs/GCE, the amount of Au is increased after gold staining.

### 3.2. LSASV Analysis of As(III)

First, the response of Au_s_/Py/C-MWCNTs/GCE to As(III) was investigated by CV and LSASV. Only the reduction peak of AuO_x_ was observed in the solution without As(III) (red line), as shown in Figure 5A. When As(III) was added to the solution (blue line), Au_s_/Py/C-MWCNTs/GCE exhibited characteristic oxidation peaks at 0.25 V and 1.00 V, which are assigned to the electrooxidation of As(0) to As(III) and then to As(V), respectively. The oxidation peak of ca. 1.10 V is due to the oxidation of Au(0) and probably H_2_O. The reduction peaks of AuO_x_ are not very different whether or not As(III) is present, indicating that the enrichment and dissolution of As(0) have little effect on the interface, which also endows the electrode with the ability of continuous detection. The analytical performance comparison of Au_/GCE_, Au_s_/C-MWCNTs/GCE, Au_s_/Py/GCE, and Au_s_/Py/C-MWCNTs/GCE for As(III) is shown in Figure 5B. Au_s_/Py/C-MWCNTs/GCE showed the highest ASV peaks of As(0) to As(III) and As(III) to As(V), and the peak shapes are also good. The Au_s_/Py/C-MWCNTs/GCE has the largest *R*_f_ value of Au, the highest enriched As(0) efficiency, and the highest analytical sensitivity, due to the largest surface area of Au on this electrode. In addition, the Au film is more stable due to the introduction of Py-functionalized C-MWCNTs on the electrode surface, which is beneficial for the stability and reproducibility of the electrode in As(III) electroanalysis.

Subsequently, the conditions for the detection of As(III) by LSASV were optimized. As shown in Appendix A, the dissolution peak heights of As(0) to As(III) and As(III) to As(V) increase with the increase in potential scan rate. However, as the sweep speed increases, the background current and noise also increase. The signal-to-noise ratio (*S*/*N*), being the ratio of current (*S*) to noise (*N*) at the dissolution peaks of As(0) to As(III) and As(III) to As(V), increases when the potential scan rate is increased. The potential scan rate of 5 V s^−1^ gave the maximum *S*/*N* ratio and was thus selected in the subsequent experiments. In addition, a high scan rate can not only save detection time, but also reduce the interference from some kinetically sluggish substances in the solution [39].

The deposition potential (*E*_D_) and deposition time (*t*_D_) were also optimized, as shown in Appendix A. As shown in Appendix A, the ASV peak heights of As(0) to As(III) and As(III) to As(V) increased as the deposition potential shifted negatively from −0.10. When the deposition potential reached −0.40 V, the signals of As(0) to As(III) and As(III) to As(V) reached saturation. Therefore, *E*_D_ is selected as −0.40 V. As shown in Appendix A, the ASV peaks of As(0) to As(III) and As(III) to As(V) increased with the increase in *t*_D_. When *t*_D_ reached 7 min, the signals of As(0) to As(III) and As(III) to As(V) reached saturation. Therefore, *t*_D_ is selected as 7 min.

The performance of Au_s_/Py/C-MWCNTs/GCE for the determination of As(III) was investigated under optimal experimental conditions. The continuous LSASV response curves of As(III) at different concentrations and the corresponding standard curves are shown in Figure 6. When the concentration of As(III) in the detection system increases, both the amount of As(0) enriched on the electrode surface and the total force of the interaction between Au and As(0) increase. Hence, the reaction of As(0) to As(III) oxidation requires more energy and gives the slight positive shift of the peak potential. The ASV peak currents of As(0) to As(III) and As(III) to As(V) have a good linear relationship with the concentration of As(III) from 0.01 to 6.00 μM (*R*^2^ = 0.997) with a sensitivity of 0.741 mA μM^−1^ and a limit of detection (LOD) of 3.3 nM (0.25 ppb) (*S*/*N* = 3), and from 0.01 to 6.00 μM (*R*^2^ = 0.991) with a sensitivity of 0.175 mA μM^−1^ and an LOD of 16.7 nM (1.20 ppb) (*S*/*N* = 3), respectively. The comparison of analytical performance in detecting As(III) with reported gold-modified electrodes is listed in Table 1. The Au_s_/Py/C-MWCNTs/GCE gives high sensitivity and a low detection limit, which can meet the requirements for the detection of As(III) in the environment. In addition, it is noteworthy that Au_s_/Py/C-MWCNTs/GCE shows a wide linear concentration range, which is due to the large *S*_Au_ that makes the enrichment of As(0) not easily reach saturation.

The stability and reproducibility of the modified electrodes were investigated. As shown in Appendix A, for the same Au_s_/Py/C-MWCNTs/GCE electrode performing five consecutive LSASV responses to 1.0 μM As(III) solution, the relative standard deviations (RSDs) of As(0) to As(III) and As(III) to As(V) peak currents are 3% and 2%, respectively. As shown in Appendix A, for the LSASV responses of five Au_s_/Py/C-MWCNTs/GCE electrodes fabricated in the same batch to 1.0 μM As(III) solution, the RSDs of As(0) to As(III) and As(III) to As(V) peak currents are 4% and 2%, respectively. After the same Au_s_/Py/C-MWCNTs/GCE electrode was stored in a refrigerator for 7 days, its detection performance was still good, and the ASV peak signal of As(0) to As(III) and As(III) to As(V) still retained 90% performance. The above results indicate that the Au_s_/Py/C-MWCNTs/GCE has good stability and reproducibility.

The interference of Cu^2+^ and As(V) in the system was investigated. In the actual detection process, Cu^2+^ that may exist in the water will form an intermetallic compound [23,43] with the detected target, which will interfere with the detection results of anodic stripping voltammetry. Thus, it is necessary to study the effect of Cu^2+^ on the detection of As(III). As shown in Appendix A, when the Cu^2+^ concentration reaches 1.0 μM, there is little effect on the dissolution peaks of As(0) to As(III) or As(III) to As(V) in 1.0 μM As(III). Therefore, there is no need to worry about the influence of Cu^2+^ in the actual sample detection. Because the deposition potential during detection is −0.40 V, while the reduction deposition of As(V) requires a more negative potential, so As(V) has little effect on the dissolution peaks of As(0) to As(III) and As(III) to As(V).

Under optimal conditions, analysis of As(III) in actual water samples (tap water, Xiangjiang River water, and Yuelu Mountain spring water) was performed on Au_s_/Py/C-MWCNTs/GCE. Analyzed by the standard addition method, the spiked water samples were filtered through a 0.22 μm filter membrane and mixed with an equal volume of 0.2 M H_2_SO_4_. The results are shown in Table 2. The good recovery values indicate the application potential of the developed electrode for As(III) analysis in actual water samples.

## 4. Conclusions

By cast-coating multiwalled carbon nanotubes on GCE and then modifying pyridine and AuNPs and staining with Au, we have prepared a Au_s_/Py/C-MWCNTs/GCE with a high surface area of Au for the sensitive and selective detection of trace As(III). To our knowledge, this is the first example of the combination of pyridine and AuNPs with carbon nanotubes for the detection of As(III), and the combination of carbon materials and organic ligands can improve the loading efficiency of AuNPs for gold staining. In this work, two-step oxidation peaks of As(0)–As(III) and As(III)–As(V) were simultaneously obtained by using fast-speed LSV to achieve the sensitive dual-signal detection of As(III). High analytical performance in the detection of As(III) was obtained. The electrode-preparation strategy may be extended to the field of noble metal electrocatalysis and electrochemical determination of drugs.

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
