# Peer review of "Anodic Stripping Voltammetric Analysis of Trace Arsenic(III) on a Au-Stained Au Nanoparticles/Pyridine/Carboxylated Multiwalled Carbon Nanotubes/Glassy Carbon Electrode"

_nanomaterials, 2022, doi:10.3390/nano12091450_

Round 1

Reviewer 1 Report

In this paper, a method for the determination of traces As(III) by ASV involving the use of a glassy carbon electrode modified with Au-stained Au nanoparticles, pyridine and carboxylated MWCNTs is proposed. It is a good article where authors have prepared an original platform which seems to work well providing good sensitivity for target detection. However, many articles have already published on the same analytical subject. Authors should justify the need for a new eIectrochemical method for As(III) and highlight the advantages of this proposal  compared to what has already been established. Iconsider the paper as publishable after major revision.

Some specific comments are as follows:

  1. Authors should justify the procedure followed for preparation of the modified electrode. Why do they use 4-cyanopyridine? Why incorporate gold nanoparticles in two steps?
  2. Figure 6. There is a slight shift in the potential of the first peak. Please, explain.
  3. Calibration plots. The units of current and concentration must be included in the equation. Also add the error of the slope and the intercept.
  4. Why haven't they used a more sensitive technique like DPV?
  5. The selectivity study should be extended to more species potentially interfering.
  6. The analyzed samples are spiked samples. Samples must be spiked prior to any pretreatment. Furthermore, the added concentration is higher than what can be expected in the water samples. Other more relevant samples, such as wastewater or even biological samples, such as urine, should be analyzed. In addition, a reference method to validate the results should be applied.

Author Response

                            Response to Reviewer 1 Comments

Point 1: Authors should justify the procedure followed for preparation of the modified electrode. Why do they use 4-cyanopyridine? Why incorporate gold nanoparticles in two steps?

Response 1: Thanks a lot for your kind advice. The use of 4-cyanopyridine (Py) can lead to the cleavage of the C-C bond between the pyridyl and the cyano group when the 4-cyanopyridine undergoes electroreduction on the electrode, resulting in a pyridine radical. The pyridine radicals are highly active and can be well covalently bound onto the surface of C-MWCNTs/GCE to form Py/C-MWCNTs/GCE. Py/C-MWCNTs/GCE can well adsorb AuNPs due to the Au-N bond. The AuNPs in this step can be used as the seeds for the next gold-staining on the electrode surface, yielding an Aus/Py/C-MWCNTs/GCE. The Au amount of Aus/Py/C-MWCNTs/GCE can be well modulated to give high analytical performance for As(III) analysis. (Please see the attachment)

Point 2: Figure 6. There is a slight shift in the potential of the first peak. Please, explain.

Response 2: Thanks a lot. As shown in Figure 6, cyclic voltammetry was used to detect the concentration of As(III). When the concentration of As(III) in the detection system increases, both the amount of As(0) enriched on the electrode surface and the total force of the interaction between Au and As(0) increase [J. Phys. Chem. C, 119 (2015) 11400-11409]. Hence, the reaction of As(0)→As(III) oxidation requires more energy and gives the slight positive shift of the peak potential. (Please see the attachment)

Point 3: Calibration plots. The units of current and concentration must be included in the equation. Also add the error of the slope and the intercept.

Response 3: Thank you very much, done so. (Please see the attachment)

Point 4: Why haven't they used a more sensitive technique like DPV?

Response 4: Thanks a lot for your kind advice. In this work, two-step oxidation peaks of As(0)-As(III) and As(III)-As(V) were simultaneously obtained by using the fast speed LSV to achieve the sensitive dual-signal detection of As(III), because the two oxidation processes are relatively facile in kinetics. The sensitivity and selectivity can be improved by the use of fast speed LSV. On the contrary, it cannot be so convenient and precise to adjust the scan rate in DPV, so DPV is not selected. (Please see the attachment)

Point 5: The selectivity study should be extended to more species potentially interfering.

Response 5: Thanks a lot. The detection of As(III) will be indeed interfered by some coexisting species in water, but copper ions are the most prominent interference among all interfering ions. The content of copper ions is usually high in aqueous solution, and the anodic peak potentials of Cu(0)-Cu(II) and As(0)-As(III) are close to each other. The anodic oxidation peak potential of other common interfering ions is quite different from that of As(0)-As(III), so the study of the interference of copper ions is the most necessary. Because the editor asked us to revise the manuscript according to the referees' comments and upload the revised file within 7 days. We are sorry that conducting the further experimental work is hard now, due to the hard situation of the COVID-19 epidemic now in our city. We will increase the exploration of possible interfering species in the future.

Point 6: The analyzed samples are spiked samples. Samples must be spiked prior to any pretreatment. Furthermore, the added concentration is higher than what can be expected in the water samples. Other more relevant samples, such as wastewater or even biological samples, such as urine, should be analyzed. In addition, a reference method to validate the results should be applied.

Response 6: Thank you very much. It is indeed an oversight of our work that the spiked concentration exceeds the expected concentration. We have supplemented the detection of real water samples according to your suggestion. To some extent, the presented standard addition analysis results have demonstrated the application potential of the developed electrode for As(III) analysis in actual water samples. Because the editor asked us to revise the manuscript according to the referees' comments and upload the revised file within 7 days. We are sorry that conducting the further experimental work is hard now, due to the hard situation of the COVID-19 epidemic now in our city. We will extend this work to more relevant samples and apply more reference methods for comparative verification. (Please see the attachment)

Reviewer 2 Report

The submitted manuscript is about the development an electroanalytical method for the determination and speciation of As (III) and As (V). Numerus electroanalytical method for the As determination has been already developed. But the submitted manuscript introduce a new modified electrode and made an appropriate characterizations that were required and is also well written. Moreover, the applied method has a good sensitivity. Therefore I recommend the publication in the current version. 

Author Response

Thanks a lot for your kind advice. Based on your comments, we have made changes to the grammar in the article.

Round 2

Reviewer 1 Report

The authors have taken into account some of the recommendations I suggested in the first review. However, some important questions such as the study of interferences other than the copper ion and the application to more relevant samples have not been carried out. The authors attribute the impossibility of carrying out these experiments to the Covid pandemic.